# The Ability of Biodegradable Thermosensitive Hydrogel Composite Calcium-Silicon-Based Bioactive Bone Cement in Promoting Osteogenesis and Repairing Rabbit Distal Femoral Defects

**DOI:** 10.3390/polym14183852

**Published:** 2022-09-15

**Authors:** Chao Guo, Junqiang Qi, Jia Liu, Haotian Wang, Yifei Liu, Yingying Feng, Guohua Xu

**Affiliations:** 1Second Affiliated Hospital of Navy Medical University, Shanghai 200003, China; 2Naval Medical Center, Shanghai 200433, China

**Keywords:** hydrogel, calcium-silica-based bone cement, anti-washout property, injectability, osteogenicity

## Abstract

Osteoporotic vertebral compression fractures are a global issue affecting the elderly population. To explore a new calcium silicate bone cement, polylactic acid (PLGA)–polyethylene glycol (PEG)–PLGA hydrogel was compounded with tricalcium silicate (C_3_S)/dicalcium silicate (C_2_S)/plaster of Paris (POP) to observe the hydration products and test physical and chemical properties. The cell compatibility and osteogenic capability were tested in vitro. The rabbit femoral condylar bone defect model was used to test its safety and effectiveness in vivo. The addition of hydrogel did not result in the formation of a new hydration product and significantly improved the injectability, anti-washout properties, and in vitro degradability of the bone cement. The cholecystokinin octapeptide-8 method showed significant proliferation of osteoblasts in bone cement. The Alizarin red staining and alkaline phosphatase activity test showed that the bone cement had a superior osteogenic property in vitro. The computed tomography scan and gross anatomy at 12 weeks after surgery in the rabbit revealed that PLGA-PEG-PLGA/C3S/C2S/POP was mostly degraded, with the formation of new bone trabeculae and calli at the external orifice of the defect. Thus, PLGA-PEG-PLGA/C_3_S/C_2_S/POP composite bone cement has a positive effect on bone repair and provides a new strategy for the clinical application of bone tissue engineering materials.

## 1. Introduction

Millions of people in the world suffer from different degrees of bone defects owing to osteoporosis, tumors, infections, sports injuries, and traffic accidents [1]. However, as the self-repair and regeneration capacity of bone is limited, accumulation of a large number of bone defects may lead to insufficient self-repair of the bone, thus requiring additional clinical treatment [2]. At present, the clinical treatment methods used for bone repair mainly include autologous bone transplantation, allogeneic bone transplantation, and xenograft bone transplantation [3]. Although autologous bone transplantation is currently the best clinical solution, this method often causes complications such as long operation time, insufficient donor site, traumatic pain, nerve damage, and blood loss [4]. Allografts and xenografts in turn significantly increase the risk of immune rejection, infection, and infectious disease [5]. To overcome these limitations of natural bone grafts, new methods for bone tissue repair or new bone substitutes need to be urgently developed.

Relevant studies have shown that calcium silicate bone cement (CSC) can promote the mineralization of bone-like apatite, which showed good cytocompatibility and degradability [6]. Peng et al. proposed that culturing of human dental pulp cells in tricalcium silicate (C_3_S) extract can induce cell proliferation [7]. However, notable disadvantages of CSC are low compressive strength and short curing time. Huan et al. developed a new calcium silicate-based/calcium sulfate composite bone cement. Compared with the traditional C_3_S or dicalcium silicate (C_2_S) cement, the composite bone cement showed a shorter self-curing time and higher compressive strength [8,9].

C_3_S/C_2_S/calcium sulfate hemihydrate bone cement is a degradable self-curing material with good biological activity [10]. However, it has several important disadvantages: (1) insufficient compressive strength and inconsistency with the elastic modulus of vertebral cancellous bone; (2) poor injectability leading to solid–liquid separation during injection; (3) poor anti-washout properties resulting in component loss to washout by the blood gushing out from the vertebral body cancellous bone during filling of the vertebral body; (4) the degradation rate in vivo is not similar to that of the cancellous bone [11,12,13].

At present, there are few reports on the application of inorganic bone cement with the liquid phase based on polyethylene glycol (PEG)/degradable polyester copolymer thermosensitive hydrogel in bone defect repair, which has a satisfying curing ability, mechanical properties, and high injectability, improving some defects of calcium silicate biological bone material such as poor mechanical properties and lower cell activity [14]. If the temperature-sensitive hydrogel is used as the liquid phase component of the inorganic bone cement, after the material is implanted into the human body, the material would self-assemble to form a gel network through hydrophobic interaction at the body temperature. This network would effectively enhance the ductility and anti-washout properties of the inorganic bone cement material in the gel bundle as well as water resistance outside the gel bundle [15,16].

In this study, polylactic acid (PLGA)-PEG-PLGA triblock polymer thermosensitive hydrogel was used as the inorganic liquid phase of bone cement, and C_3_S/C_2_S/calcium sulfate hemihydrate was used as the solid phase, thus forming a PLGA-PEG-PLGA/C_3_S/C_2_S/plaster of Paris (POP) composite bone cement system. We measured its physicochemical properties and cytotoxicity, and determined in an in vivo experiment whether PLGA-PEG-PLGA/C_3_S/C_2_S/POP can solve the problems of solid–liquid separation, easy hemodilution and collapse by cancellous bone, and mismatch between the degradation rate and the ingrowth rate of new bone during injection.

## 2. Materials and Methods

### 2.1. Preparation of PLGA-PEG-PLGA/C_3_S/C_2_S/POP

C_3_S powder was prepared using the sol–gel method [17]. C_2_S powder was synthesized using the co-precipitation method [17]. Calcium sulfate hemihydrate was prepared via drying and dehydration of calcium sulfate dihydrate [18]. C_3_S, C_2_S, and calcium sulfate hemihydrate were mixed at a mass ratio of 4:1:2, ball-milled for 12 h, and passed through a 300-mesh sieve to obtain C_3_S/C_2_S/POP composite bone cement powder. PLGA-PEG-PLGA thermosensitive hydrogel was synthesized via ring-opening polymerization [19,20]. Using PEG as the macromolecular initiator, under the protection of argon, lactide and glycolide monomers at different molar ratios were added, and then the stannous octoate catalyst was added, followed by polymerization at 150 °C. The obtained crude polymer product was washed with water and purified, freeze-dried, and stored in vacuum at −20 °C for further use. Using PEG with different chain lengths as initiators and feeding according to the number-average molecular weight ratio of PLGA/PEG of 2.5/1, block copolymers with successively increasing chain lengths were obtained, thereby yielding PLGA-PEG-PLGA thermosensitive hydrogels.

Two 2 mL syringes connected with a tube were filled with hydrogel and deionized water, and the solidification solutions for bone cement with the following volume percentages of hydrogel: 0 (further referred to as blank), 10, 20, 30, and 40%, were prepared. The C_3_S/C_2_S/POP bone cement powder was mixed with the prepared bone cement liquid phase at the liquid-to-powder ratio of 0.5 mL/g and vigorously stirred for 1 min to obtain the bone cement slurry.

### 2.2. Characterization of Physical and Chemical Properties

#### 2.2.1. Hydration Product Analysis

The obtained bone cement slurry was injected into a cylindrical mold with a diameter of 6 mm and height of 12 mm. After the final setting time elapsed, the cement cylinder was extracted from the mold using a stripper rod (Figure 1). The cement cylinder was placed in a bottle with simulated body fluid (SBF), which was placed in a 37 °C water bath for 1 d. Subsequently, the cement sample was removed from the bottle, soaked in absolute ethanol for 2 h, and dried at room temperature for 30 min. Finally, the cement cylinder was ground into powder and analyzed using X-ray diffraction.

#### 2.2.2. Setting Time

Curing time was measured using a Vicat instrument according to the Cement Curing Time Test Standard [21]. The initial and final setting needles were dropped vertically on the cement column every 1 min. The starting point for the initial and final setting time measurements was the moment of the preparation of the cement slurry. The end point of the initial setting time measurement was the moment at which the light needle stayed on the surface of the cement column for 5 s. The end point of the final setting time measurement was the moment when the heavy needle of the Vicat instrument no longer left a circular indentation on the surface of the bone cement column. Three parallel samples were tested each time, and the average value was recorded.

#### 2.2.3. Compressive Strength

The compressive strengths of the composite bone cements with different hydrogel contents and different curing times were tested. After the final setting of the prepared cement cylinder, it was placed in a SBF at 37 °C, where it was soaked for different time periods (12 h–14 d), after which it was removed from the solution and dried at room temperature for 30 min. The compressive strength of the cement cylinder was tested using a mechanical testing machine. The loading speed of the mechanical testing machine was 0.5 mm·min^−1^, and the average of three parallel tests was recorded.

#### 2.2.4. Injectability

The injectability was determined using a device shown in Figure 2. A 50 mL medical syringe with an outlet diameter of 2 mm was filled with the as-prepared bone cement slurry. The filled syringe was placed into a water bath thermostatic oscillator with a temperature of 23 °C and relative humidity of 100%. After 1 min, the cement slurry was pushed out. The weight of the empty syringe before loading was recorded as m_0_, the mass of the syringe after loading was recorded as m_1_, and the mass of the syringe after the bone cement was pushed out was recorded as m_2_. Injectability was calculated as: injectability = (m_1_ − m_2_)/(m_1_ − m_0_) × 100%. Three parallel samples were tested each time, and the average value was taken as the test result.

#### 2.2.5. Anti-Washout Properties

The as-prepared bone cement slurry was loaded into a 2.5 mL medical syringe with an outlet diameter of 2 mm. The filled syringe was put into a water bath thermostatic oscillator with a temperature of 3 °C and relative humidity of 100% for 5 min. Then, the cement slurry was continuously pushed out into a glass dish with SBF at 37 °C, which was shaken for 1 min using a shaker. The integrity and water resistance of the bone cement were observed after shaking.

#### 2.2.6. In Vitro Degradation

The samples used in degradability tests were composite bone cement discs with a diameter of 6 mm and a height of 2 mm. First, the prepared cement discs were dried and then weighed on a four-position balance (recorded as m_0_); then, the cement discs were immersed in the SBF at the ratio of the specific surface area of the cement discs to the SBF volume of 0.1 cm^2^/cm^3^. After reaching the preset time, the sample was taken out, rinsed with deionized water, dried in a 60 °C oven for 24 h, and weighed (recorded as m_n_). After weighing, the samples were subjected to the same treatment with fresh SBF, and dried and weighed using the same method as above at subsequent time points. The degradation rate of the material was calculated as: degradation rate = (m_0_ − m_n_)/m_0_ × 100%. Three parallel samples were tested each time, and the average value was taken as the test result.

### 2.3. In Vitro Experimental Studies

#### 2.3.1. In Vitro Cell Compatibility Evaluation

MC3T3-E1 mouse osteoblasts (Bohu Biotechnology Co., Ltd., Shanghai, China) were cultured and heated on PLGA-PEG-PLGA/C_3_S/C_2_S/POP (0, 10, 20, 30, and 40% *v*/*v*) columns, and then analyzed using a cholecystokinin octapeptide (CCK)-8 kit (Dojindo, Kumamoto, Japan). The relative growth rate (RGR) of the cells was calculated from optical density (OD), using the following formula: RGR = (OD of the experimental group/OD of the blank control group) × 100%. If RGR > 75%, the cell compatibility of the material was considered good, and if RGR > 100%, the material was considered to promote cell proliferation.

#### 2.3.2. Alizarin Red Staining and Quantitative Detection

Two parallel samples (composite bone cement columns) for each hydrogel ratio of the PLGA-PEG-PLGA/C_3_S/C_2_S/POP were ground into powders. The mixed culture liquid was sterilized at a concentration of 10 mg/mL. The stem cells were seeded in 24-well plates. After 24 h, the old culture medium was removed, and 0.5 mL of osteogenic induction medium (Sigma, Hiroshima, Japan) was added. Induction was performed for 14 d. The cells were then fixed for 15 min with 4% paraformaldehyde (Sigma, Japan) and stained with Alizarin Red solution (Beyotime, Petaluma, CA, USA) for 3 min, after which they were observed under a microscope and photographed. Subsequently, hexadecyl chloride was added and left at room temperature for 30 min, the supernatant was drawn, and the absorbance of the sample was measured at 560 nm using a microplate reader (Biotech, Shanghai, China), while simultaneously measuring the absorbance of the supernatant of a group of simple cells. The difference between the absorbances of the sample and the simple cell groups was recorded.

#### 2.3.3. Quantitative Detection of Alkaline Phosphatase (ALP) Activity

Grouping and culturing were performed as described above. The cells were lysed (Beyotime, Petaluma, CA, USA), the supernatant was taken, and the ALP kit (Beyotime, Petaluma, CA, USA) was used to test samples. The absorbance of each sample was examined thrice, and the average value was recorded.

### 2.4. Establishment of Animal Models and Material Implantation

Animal experiments were approved by the Experimental Animal Ethics Committee of Shanghai Jiaotong University (No. 20191244). Eighteen female experimental New Zealand white rabbits with established early-stage osteoporosis models were divided into three groups (A, B, and C; 6 in each group): group A was the experimental group (30% PLGA-PEG-PLGA/C_3_S/C_2_S/POP composite bone cement injection group), group B was the polymethyl methacrylate (PMMA) control group (PMMA bone cement injection group), and group C was the control group. After the anesthesia took effect, the distal femur was prepared, and the rabbit was fixed in the supine position on the animal operating table. The slightly raised platform structure was palpable on the lateral side of the femur 1 cm above the knee joint, and the position of the lateral femoral condyle was determined. A longitudinal incision of approximately 2.5 cm in length was made in the axial direction, and the subcutaneous tissue, muscle, and periosteum were separated layer by layer, taking care not to damage the common peroneal nerve. A cylindrical bone defect area with a diameter of 6 mm and a depth of 6 mm was drilled at the lateral femoral condyle using a low-speed limited deep trephine drill, and the bone debris and blood accumulation in the defect were thoroughly washed with normal saline. Depending on the experimental group, 30% PLGA-PEG-PLGA/C3S/C2S/POP, PMMA, or nothing was implanted. The incision was sutured conventionally and fixed with a dressing.

At 6 and 12 weeks after surgery, 3 rabbits in each group were sacrificed through air embolization. These rabbits were entered from the original surgical approach, and the distal femur surgical site specimens were harvested for imaging.

### 2.5. Statistical Analysis

Data are presented as the means ± standard deviations. Statistical difference was analyzed by SPSS software using one-way analysis of variance (ANOVA) with Tukey’s test, with *p* < 0.05 considered a significant level.

Appendix A shows the experimental flow chat in the present study.

## 3. Results

### 3.1. Physical and Chemical Properties of the PLGA-PEG-PLGA/C_3_S/C_2_S/POP Composite Bone Cement

The phase characterization results of the hydration products of the PLGA-PEG-PLGA/C_3_S/C_2_S/POP composite bone cement are shown in the XRD pattern in Figure 3: the addition of hydrogel does not lead to the formation of new phases.

The results of curing time tests (Figure 4) show that the addition of degradable polyester hydrogel increases the curing time of CSC. The initial setting time and final setting time of blank C_3_S/C_2_S/POP bone cement are 14.3 and 25.5 min, respectively. At 23 °C, the curing times of the samples containing hydrogel are significantly higher than that of the blank C_3_S/C_2_S/POP group (*p* < 0.05). In the 37 °C water bath, the curing times are significantly shorter, which was significantly lower than that of the corresponding group at 23 °C (*p* < 0.05).

The compressive strength of the composite bone cement with different hydrogel contents (liquid-to-powder ratio was the same for all samples) increases with curing time and reaches the maximum on the 14th day. With the increase in hydrogel content, the compressive strength of bone cement also gradually increases. At 30 and 40% PLGA-PEG-PLGA, the compressive strengths on the 14th day are 45.6 ± 6.6 and 44.4 ± 8.0 MPa, which are significantly different from that of the blank C_3_S/C_2_S/POP cement on the 14th day (*p* < 0.05). At the same time, the compressive strengths of the 10 and 20% PLGA-PEG-PLGA groups are not significantly different from that of the control group (Figure 5).

Figure 6 shows that the injectability of the composite bone cement significantly increases with the content of thermosensitive hydrogel. The 30% PLGA-PEG-PLGA/C_3_S/C_2_S/POP composite bone cement exhibits a high injectability of 85.9 ± 3.5%, which is significantly higher than that of the blank bone cement (56.6 ± 3.5%, *p* < 0.05). However, with the further increase in the hydrogel content, the injectability of the composite bone cement decreases: the injectability of the 40% PLGA-PEG-PLGA/C_3_S/C_2_S/POP composite bone cement is 78.1 ± 4.2%, which is still significantly higher than that of the blank group (*p* < 0.05).

Figure 7 shows the test simulating the injection of bone cement into cancellous bone, where the cement may be washed out by blood. The as-prepared bone cement slurry was directly injected from a syringe into the 37 °C SBF, and the anti-washout properties of each group were observed. Figure 7 shows that the blank bone cement without hydrogel has poor anti-washout properties because it exhibits considerable washout after the 1 min shaking test. However, the 30 and 40% PLGA-PEG-PLGA/C_3_S/C_2_S/POP composite bone cements show good anti-washout properties. The injected bone cement slurry in these groups is continuous and maintains integrity by the end of the test.

The degradation of each group of bone cement in buffer solution is shown in Figure 8, where two tendencies can be clearly observed. First, the weight loss percentage for all materials gradually increases with soaking time; second, at the same soaking time, the weight loss percentage increases with the increase in hydrogel content. In the 10th week, the degradation rates in the 30 and 40% groups are 46.3 ± 4.4 and 48.3 ± 5.2%, respectively, which are significantly higher than that of blank bone cement (*p* < 0.05).

### 3.2. Cell Proliferation Activity and Osteoinductive Properties of the PLGA-PEG-PLGA/C_3_S/C_2_S/POP Composite Bone Cement

Cell proliferation activity was detected using CCK-8 colorimetry. The results show that the cell proliferation ability of the PLGA-PEG-PLGA/C_3_S/C_2_S/POP composite bone cement increases with the PLGA-PEG-PLGA hydrogel content (Figure 9). To further compare the cell proliferation of each cement group, the concept of the “relative growth rate” was used. In all groups, the relative growth rate is above 95% (Figure 10). The cell growth rate of the 40% PLGA-PEG-PLGA/C_3_S/C_2_S/POP on the third day is comparable to that of blank C_3_S/C_2_S/POP bone cement. The relative growth rate of cement is significantly different (*p* < 0.05).

Figure 11 shows the macroscopic photos of the composite bone cement samples co-cultured with mouse embryonic osteogenic precursor cells for 14 d and stained with alizarin red. After 14 d, the staining degrees of 20–40% PLGA-PEG-PLGA/C3S/C2S/POP samples are significantly higher than that of the blank C_3_S/C_2_S/POP bone cement. The quantitative results of the test are shown in Figure 12. After 14 d, the PLGA-PEG-PLGA/C_3_S/C_2_S/POP groups all show higher ODs than the blank bone cement. The OD of the 30% PLGA-PEG-PLGA/C3S/C2S/POP group is significantly higher than that of blank cement (*p* < 0.05). Therefore, with the increase in the PLGA-PEG-PLGA hydrogel content, the OD first increases and then decreases, with the highest activity observed for 30% PLGA-PEG-PLGA/C_3_S/C_2_S/POP.

The results of the ALP activity test for each group are shown in Figure 13. After 14 d, ALP activities of bone cements with 10–40% PLGA-PEG-PLGA contents are higher than that of the C_3_S/C_2_S/POP group. Notably, the ALP activity of the 30% PLGA-PEG-PLGA is significantly higher than those in other groups (*p* < 0.05).

### 3.3. Animal Experiment

The average duration of the operation on experimental animals was 21.8 ± 4.1 min, and they woke up about 1 h after the operation. Two days after the operation, the spirit, diet, and bowel movements returned to normal. There was no infection in the three groups of animals after the operation, no redness, swelling, oozing, sinus tract formation, or endocrine formation at the incision, and the incision healed in about 10 d. There was a decreased range of motion, fractures, and other symptoms.

Six weeks after the operation, experimental specimens from rabbits were examined using micro-computed tomography (CT) (Figure 14). In group A (30% PLGA-PEG-PLGA/C_3_S/C_2_S/POP), the implanted bone cement material is evenly distributed in the cancellous bone, and a small amount of bone cement penetrates into the small bone. Inside the beam, the defect site is fully filled with no evident calluses formed. In group B (PMMA), the implanted bone cement material is evenly distributed in the cancellous bone, but there is a small amount of bone tissue necrosis around it, which is caused by the heat generation and monomer toxicity of the PMMA bone cement. In group C (control), the bone defect is still evident, with no notable bone regeneration observed, and there are more bone defects at the outer mouth.

Micro-CT images of the experimental specimens of the rabbits 12 weeks after the operation are shown in Figure 15. In group A, the bone cement has an irregular shape, and it is mostly degraded, absorbed, and surrounded by new bone tissue, without trabecular damage; moreover, the defect is replaced by new cancellous bone. Compared with the image at 6 weeks (Figure 14), the bone hyperplasia and bone cement absorption are more pronounced. In group B, the bone cement is not absorbed, the boundary is clear, no new bone formation around is observed, and the bone cement density is significantly different from that of the surrounding bone tissue. In group C, the formation of a small amount of trabecular bone is observed, the outer opening of the defect is not completely closed, and the bone defect is still visible in the central area.

Figure 16 shows the lateral view of the femoral condyle for the three groups of animals at 12 weeks after the operation. In the PLGA-PEG-PLGA/C_3_S/C_2_S/POP group, the bone defect area is completely healed, and a small amount of callus formed around the external port is observed.

## 4. Discussion

In general, the hydration of C_3_S and C_2_S to xCaO·SiO_2_·yH_2_O (x = 1.2–2.3) is slow [22,23,24]. The phase analysis of hydration products formed after 24 h of hydration curing shows that unconverted C_3_S and C_2_S are still present. At the same time, the hydration of calcium sulfate hemihydrate yields dihydrate gypsum, which has a rod-like morphology and forms an interlaced and overlapping porous structure [25,26]. Thus, PLGA-PEG-PLGA hydrogel was compounded with C_3_S/C_2_S/POP in the present study to explore a new type of CSC in bone repair.

The cross-linking reaction of the introduced thermosensitive hydrogel results in a three-dimensional cross-network structure [27], thereby increasing the bonding strength between C_3_S/C_2_S/POP, reducing the porosity, and increasing the compressive strength of PLGA-PEG-PLGA/C_3_S/C_2_S/POP to a certain extent. The swelling and release profiles of the thermosensitive hydrogels can be tailored by varying the balance of hydrophilicity and hydrophobicity of the network structures and the crosslinking density [28]. At the same time, the degradable polyester polymer in the liquid phase of the bone cement forms a network hydrogel when it encounters Ca^2+^, and the chelation effect is also beneficial for the improvement in the compressive strength. The experimental results after 14 d of curing show that the compressive strengths of 30 and 40% PLGA-PEG-PLGA/C_3_S/C_2_S/POP are significantly higher than that of the blank cement. However, at the temperature-sensitive hydrogel content of 40%, the injectability of the composite bone cement significantly decreases compared to that at 30%. This is because, at the constant liquid–solid ratio, a high hydrogel content significantly reduces the amount of water in the liquid phase, thereby affecting the mixing of bone cement powder and water. As a result, a large proportion of particles remain unreacted, reducing the injectability [20]. Usually, the needle part of the syringe used in clinical applications is small in diameter and has considerable length. Thus, during bone cement injection, the particles may agglomerate at the catheter port and be difficult to push out [29]. When excessive pressure is applied, solid–liquid separation greatly reduces the injection performance. The introduction of thermosensitive hydrogel polymer significantly improves the injectability of bone cement, which is attributed to the change in bone cement viscosity [30,31]. The viscosity of the slurry affects the flow performance and injectability of the system. The addition of high-molecular polymers to the bone cement increases the viscosity of the bone cement liquid phase because of the high viscosity of the polymer after curing. The liquid phase with higher viscosity is less prone to agglomeration and blockage of bone cement particles at the nozzle of the syringe, thereby improving the injectability [32,33]. The C_3_S/C_2_S/POP bone cement shows poor anti-washout performance, and during injection, it is easily dispersed by the blood in the vertebral body [34,35]. The results of this experiment show that the introduction of PLGA-PEG-PLGA thermosensitive hydrogel polymer can significantly improve the anti-washout performance of C_3_S/C_2_S/POP bone cement, especially at 37 °C, where the hydrogel changes from a liquid to solid state and considerably increases anti-washout properties. In addition, the introduction of degradable polyester polymer was found to extend the curing time of CSC. At the same time, at 37 °C, the curing time was significantly shorter, and there were no significant differences with the control group. Overall, the obtained results meet the clinical requirements.

Bone cement used in the human body must have good biocompatibility, which is not only conducive to the repair and balance of biological tissues but also conducive to the performance of implant materials [36,37]. According to the literature, both C_3_S/C_2_S/POP and PLGA-PEG-PLGA have good biocompatibility [38,39]. However, some studies suggest that biological substitute materials may excessively increase the reactive oxygen species (ROS) of osteoblasts around the material, which inhibits the activity of osteoblasts and is detrimental to the repair of surrounding bone [20]. The PLGA-PEG-PLGA hydrogel not only has physical and chemical properties similar to the human body environment but also scavenges oxygen free radicals, which promotes osteogenesis [40]. Experiments show that the addition of PLGA-PEG-PLGA can not only prevent the excessive increase in intracellular ROS but also reduce the degree of oxidative damage to the surrounding cells to a certain extent, and promote the survival of cells [32]. The CCK-8 experiment showed that the OD of PLGA-PEG-PLGA/C_3_S/C_2_S/POP is higher than that of the control group, which not only confirmed its good biocompatibility but also indicated that the 40% PLGA-PEG-PLGA sample was the most favorable for the adhesion and proliferation of MC3T3-E1 cells on C_3_S/C_2_S/POP. The results of alizarin red staining and ALP activity tests showed that the addition of PLGA-PEG-PLGA polymer did not affect the osteogenic differentiation ability of MC3T3-E1, but at a certain concentration, it significantly increased the osteogenic performance of the material itself. Considering the polymerization, the addition of the compound increased the surface electronegativity of the C_3_S/C_2_S/POP material, which was conducive to the adhesion and proliferation of MC3T3-E1 cells on C_3_S/C_2_S/POP. Therefore, the composite PLGA-PEG-PLGA copolymer and C_3_S/C_2_S/POP have good biological activity, good osteoblast adhesion, good growth, and good osteoconductivity.

The role of bone graft substitute materials in bone repair can be divided into osteoconduction, osteoinduction, and osteogenesis. The proposed composite bone cement showed an ideal biocompatibility and osteoconductive effect, which is mainly related to the significant osteoinductive mechanism of C_3_S/C_2_S/POP [41]. In calcium-silicate-based biomaterials, the release of SiO_3_^2−^ and Ca^2+^ ions affects the cell cycle of bone-related cells and genes that regulate bone regeneration, thereby stimulating bone regeneration [42]. Under the action of these ions, genes encoding transcription factors and growth factors, such as insulin-like growth factor, are activated and up-regulated in osteoblasts, and cell cycle processes regulated by these factors, such as cell proliferation and differentiation, are enhanced [43]. Calcium-silicate-based materials have ideal biological activity and can induce bone-like apatite mineralization in the physiological environment or SBF, and thus establish chemical bonds between the material and surrounding tissues [44,45]. The introduction of the thermosensitive hydrogel accelerated the degradation rate of the composite bone cement. CSC biodegrades mainly via two pathways: passive absorption through chemical dissolution and active cell-mediated absorption [46,47]. However, complete degradation of pure calcium-silicate-based bone cement is difficult to achieve, meaning that a large amount of filling material would not be completely degraded, which is not conducive to the formation of new bone in the vertebral body. The introduction of PLGA-PEG-PLGA thermosensitive hydrogel accelerated the degradation of composite bone cement, and the degradation rate could be controlled by adjusting the hydrogel content. In this experiment, most of the implanted 30% PLGA-PEG-PLGA/C_3_S/C_2_S/POP composite bone cement degraded by 12 weeks after the operation. In the early postoperative period, the bone defects in the experimental group were filled with composite bone cement, and there was also a small amount of bone cement in the trabecular structure. During bone growth through osteoconduction, new bone tissue is formed around the cement and at the degradation sites, thereby leading to osseointegration with the cement and increasing the strength of the filling site as a whole [48,49]. However, PMMA lacks corresponding biological activity and cannot fuse with autologous bone [50]. Owing to the heat released in the early stage of solidification and mismatch with the elastic modulus of cancellous bone, a small amount of bone necrosis was observed around the PMMA filling, and the boundaries between PMMA and the surrounding bone were clear. However, no re-fractures occurred. At the same time, the defect in the blank control group was not completely healed at 12 weeks after the surgery. Thus, PLGA-PEG-PLGA/C_3_S/C_2_S/POP composite bone cement showed suitable biomechanical strength. Although its early strength is not as good as that of PMMA, over time, it is conducive to the regeneration of trabecular bone and the overall bone defect site, resulting in the continual increase in compressive strength. Therefore, the proposed composite bone cement is a promising material for spinal bone repair.

## 5. Conclusions

A new degradable polyester thermosensitive hydrogel/CSC system was designed in this study. With the intention of maintaining the good bioactivity and biocompatibility of calcium-silicate-based bone cement, the prepared composite bone cement showed better injectability, better anti-washout properties, faster biodegradability, better biocompatibility, and better matching of the elastic modulus with those of cancellous bone than pure calcium-silicate-based bone cement. Through in vitro cell experiments and animal experiments, the composite bone cement was confirmed to be safer and possess more effective osteogenic properties as a bone filling material than pure calcium-silicate-based bone cement. In the future, prospective cohort studies are required for the osteoporotic vertebral compression fractures to fully verify the effectiveness and superiority of this composite bone cement system.

## Figures and Tables

**Figure 1 polymers-14-03852-f001:**
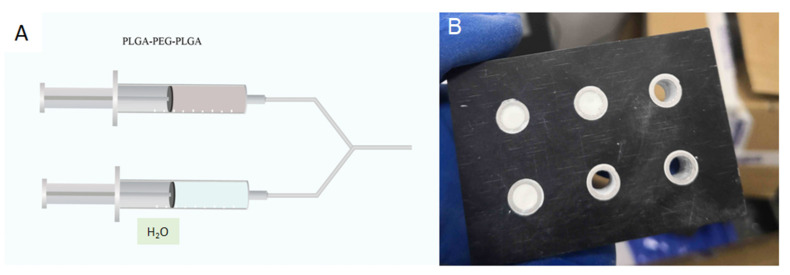
Preparation of PLGA-PEG-PLGA/C_3_S/C_2_S/POP composite bone cement column: (**A**) bone cement liquid phase mixing injection device; (**B**) PTFE cylinder mold.

**Figure 2 polymers-14-03852-f002:**
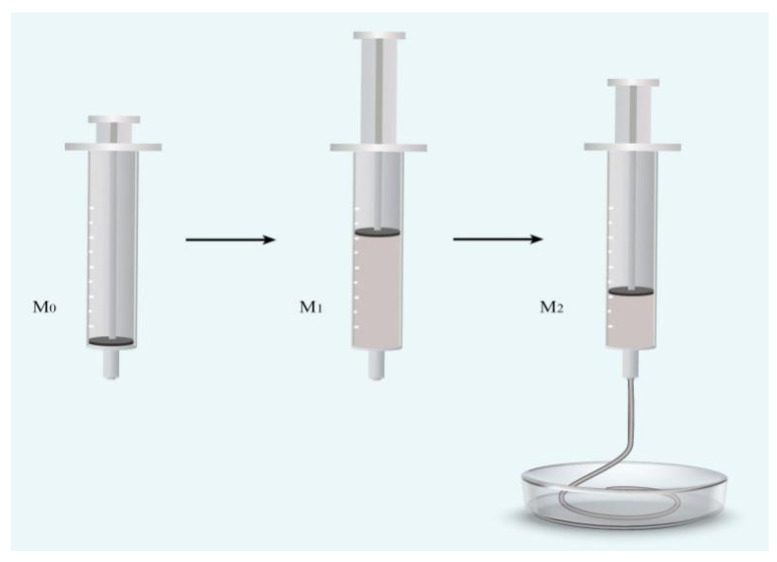
Measurement of injectability.

**Figure 3 polymers-14-03852-f003:**
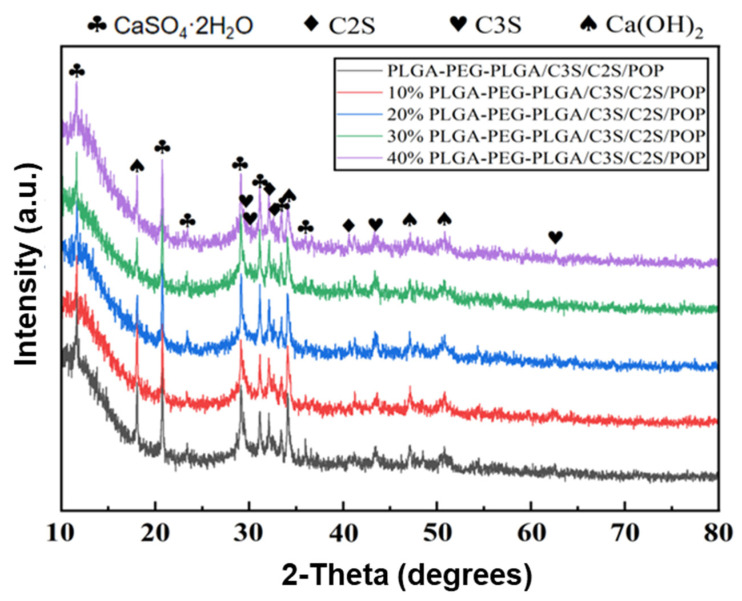
XRD pattern of hydration products of PLGA-PEG-PLGA/C3S/C2S/POP bone cement. The standard card numbers (ICDD PDF NO) corresponding to the objects shown in the figure are as follows: CaSO_4_ 2H_2_O: 33–0311, C_3_S: 49–0442, C_2_S: 33–0302, Ca(OH)_2_: 44–1481.

**Figure 4 polymers-14-03852-f004:**
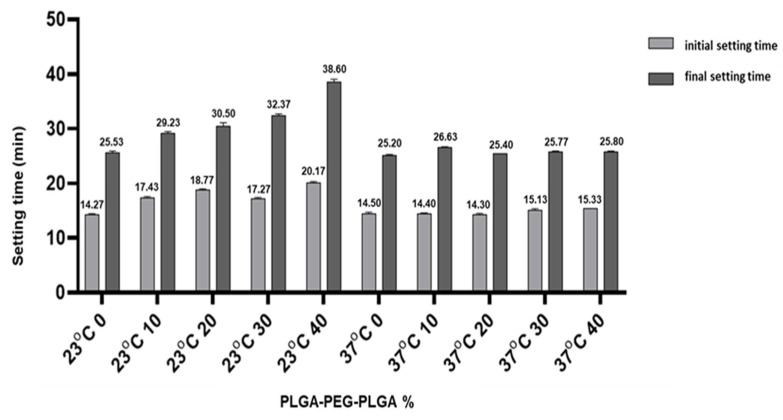
Curing time of composite bone cement with different PLGA-PEG-PLGA temperature-sensitive hydrogel contents at different temperatures.

**Figure 5 polymers-14-03852-f005:**
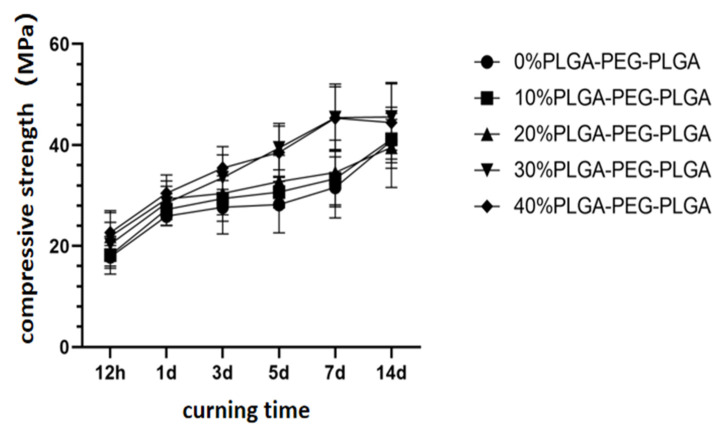
Changes in the compressive strengths of composite bone cements with different PLGA-PEG-PLGA thermosensitive hydrogel contents with curing time.

**Figure 6 polymers-14-03852-f006:**
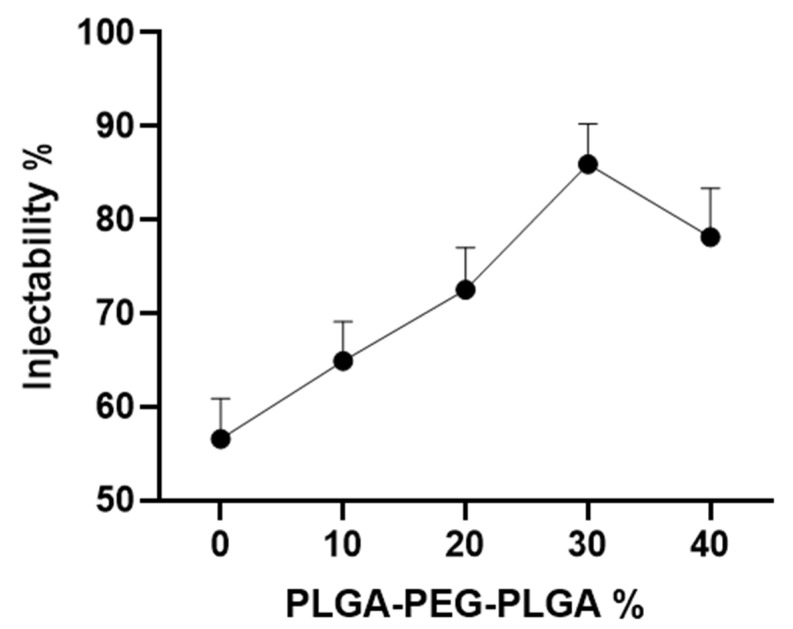
Injectability of composite bone cement with different PLGA-PEG-PLGA thermosensitive hydrogel contents.

**Figure 7 polymers-14-03852-f007:**
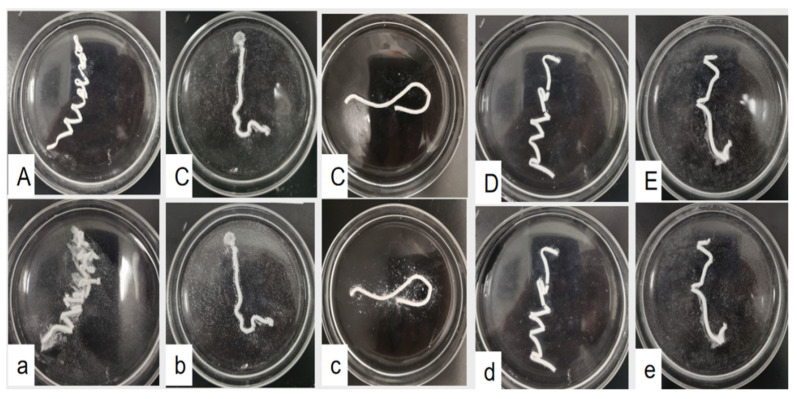
Anti-washout properties of the composite bone cement with different PLGA-PEG-PLGA temperature-sensitive hydrogel contents: (**A**–**E**) photos of as-prepared 0, 10, 20, 30, and 40% *v*/*v* PLGA-PEG-PLGA/C_3_S/C_2_S/POP bone cement slurry injected into SBF solution at 37 °C, respectively; (**a**–**e**) corresponding photos after shaking using a horizontal rotary oscillator for 1 min.

**Figure 8 polymers-14-03852-f008:**
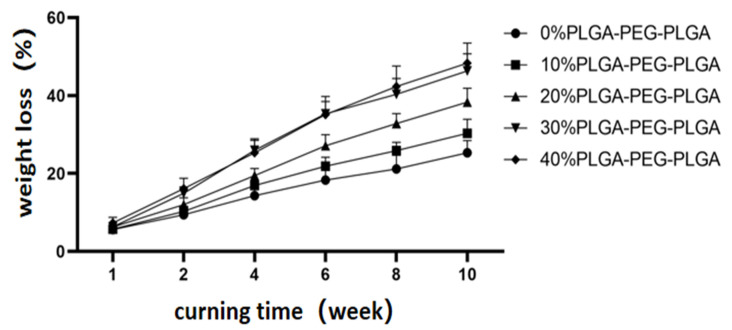
Degradation of the composite bone cements with different PLGA-PEG-PLGA temperature-sensitive hydrogel contents with curing time.

**Figure 9 polymers-14-03852-f009:**
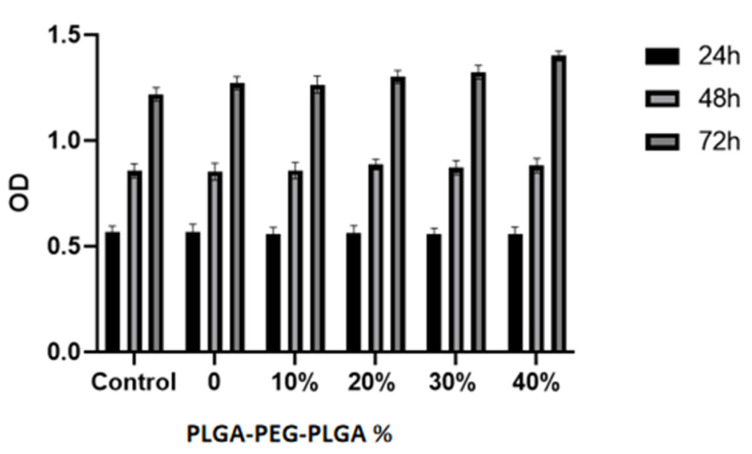
Cytocompatibility of the composite bone cements with different PLGA-PEG-PLGA thermosensitive hydrogel contents.

**Figure 10 polymers-14-03852-f010:**
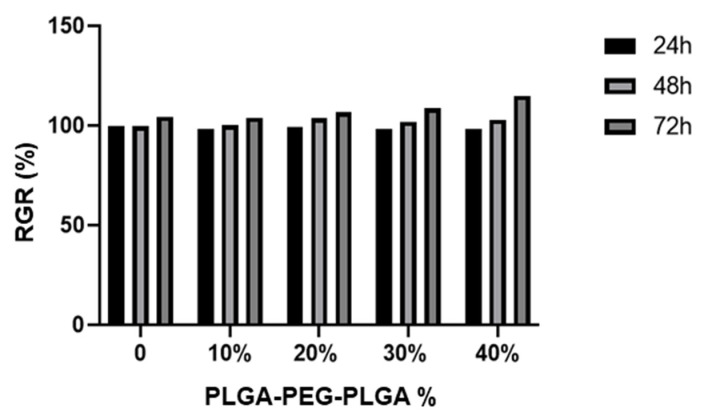
Relative growth rate of composite bone cement with different PLGA-PEG-PLGA temperature-sensitive hydrogel contents at different times.

**Figure 11 polymers-14-03852-f011:**
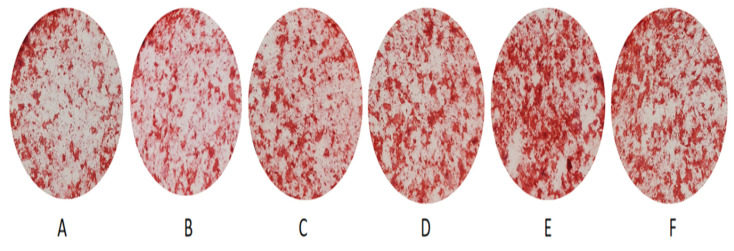
Alizarin red staining results after 14-day culture: (**A**–**F**) the control, 0, 10, 20, 30, and 40% PLGA-PEG-PLGA/C_3_S/C_2_S/POP, respectively.

**Figure 12 polymers-14-03852-f012:**
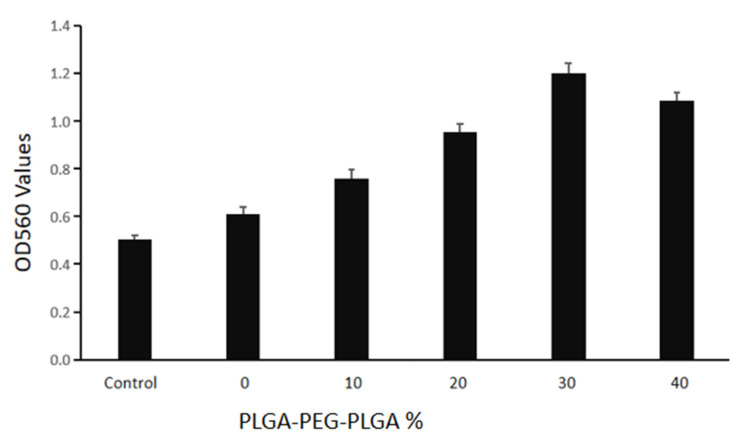
Quantitative results of alizarin red staining for each group.

**Figure 13 polymers-14-03852-f013:**
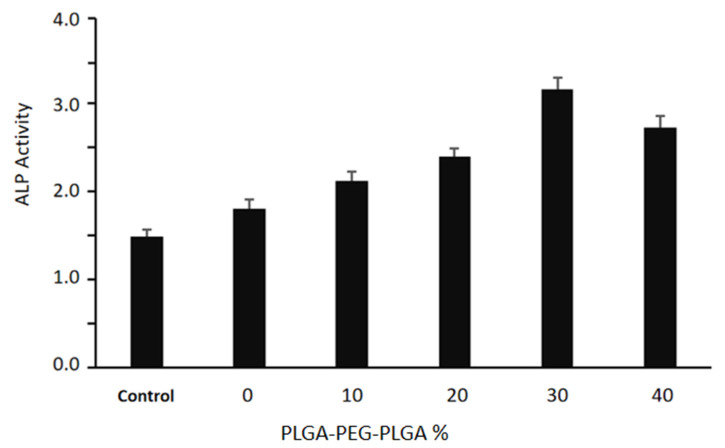
Quantitative results of ALP activity test after 14-day culture.

**Figure 14 polymers-14-03852-f014:**
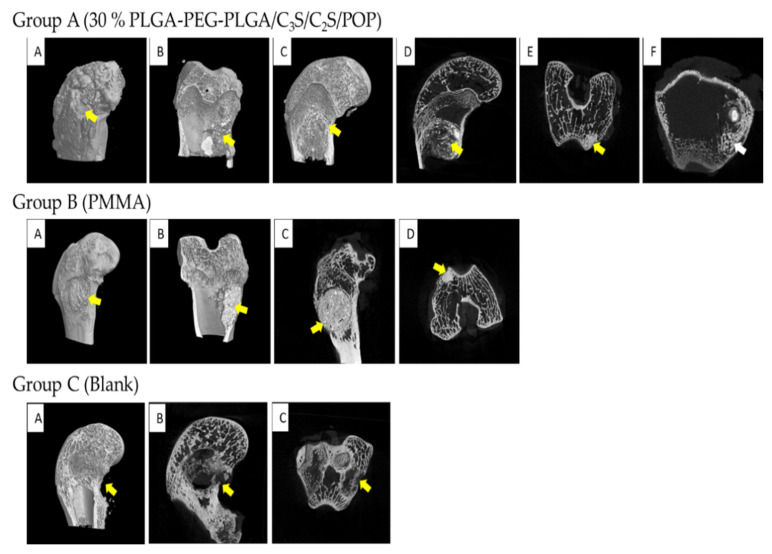
MICRO-CT results of femoral condyles and implant material 6 weeks after operation. Group A (30% PLGA-PEG-PLGA/C3S/C2S/POP): (**A**–**C**) three-dimensional reconstruction; (**D**–**F**) sagittal, coronal, and horizontal, respectively. The yellow arrow pointed to the composite bone cement filler material and the white arrow pointed to the new bone trabecula. Group B (PMMA): (**A**,**B**) reconstructed in three dimensions; (**C**,**D**) sagittal and coronal, respectively. The yellow arrow refers to PMMA bone cement filling material, which had a clear boundary with the surrounding bone, and no obvious new bone trabecula was found. Group C (Blank): (**A**) three-dimensional reconstruction map, (**B**) sagittal plane, and (**C**) coronal plane. Yellow arrow pointed to the defect of the femoral condyle, and no healing was observed.

**Figure 15 polymers-14-03852-f015:**
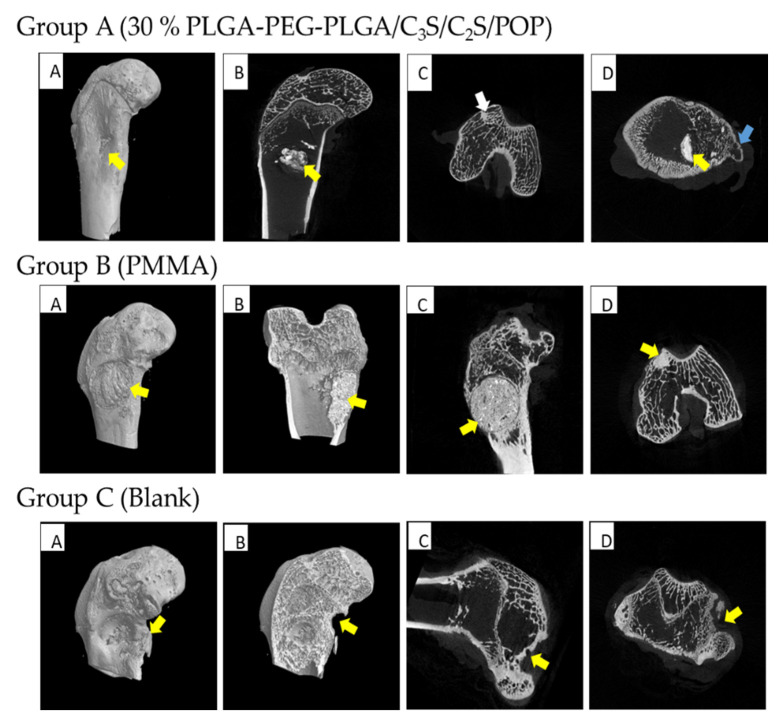
MICRO-CT results of femoral condyles and implant material 12 weeks after operation. Group A (30% PLGA-PEG-PLGA/C3S/C2S/POP): (**A**) three-dimensional reconstruction, (**B**–**D**) sagittal, coronal, and horizontal planes, respectively. The yellow arrow pointed to the composite bone cement filler, and the filler was partially degraded, so the bone defect could be completely healed. The white arrow pointed to the new bone trabecula. The blue arrow indicates a nascent callus. Group B (PMMA): (**A**,**B**) reconstructed in three dimensions, and (**C**,**D**) sagittal and coronal, respectively. The location pointed by the yellow arrow was the filling material of PMMA bone cement. The filling material did not degrade, and the boundary with the surrounding bone substance was clear. Group C (Blank): (**A****,B**) three-dimensional reconstruction map, (**C**) sagittal plane, and (**D**) coronal plane. The yellow arrow pointed to the defect of the femoral condyle, there was little generation of new bone trabeculae, and no complete healing was observed.

**Figure 16 polymers-14-03852-f016:**
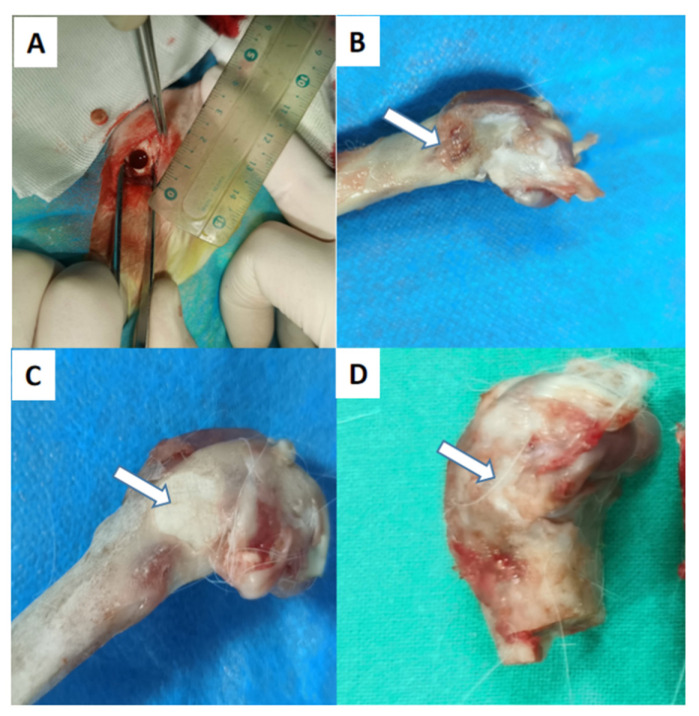
Gross lateral views of the femoral condyle 12 weeks after the operation in three groups of animals: (**A**) bone defect with a diameter of 6 mm drilled through an annulus during the operation; (**B**) blank control group; (**C**) PMMA group; (**D**) 30% PLGA-PEG-PLGA/C_3_S/C_2_S/POP group.

## Data Availability

The data presented in this study are available on request from the corresponding author. The data are not publicly available, due to the next work.

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
