# Peer review of "The Ability of Biodegradable Thermosensitive Hydrogel Composite Calcium-Silicon-Based Bioactive Bone Cement in Promoting Osteogenesis and Repairing Rabbit Distal Femoral Defects"

_polymers, 2022, doi:10.3390/polym14183852_

Round 1

Reviewer 1 Report

Below I have listed most important weaknesses of the manuscript that I have spotted during my reading.

  1. Introduction is rather modest and it does not provide with adequate background. Moreover, the review of recent findings in the field is, let me say, missing. The most recent literature entry in the bibliography was published in 2017 and the area is developing fast. I Suggest to extend the introduction in a way that build narration providing with the proper and recent background of the research based on the contemporary literature.
  2. I suggest including in the Abstract and Introduction some content of scientific significance of the study.
  3. In some places I have found the “space” missing after full sot. Please revise thoroughly all the text. I have spotted the problem at line: 33, 35, 423, 430, 435, 450, 453
  4. Some superscripts and subscripts are not formatted see lines: 120, 147, 447.
  5. The degree symbol ( ° ) is missing or replaced with some strange characters see lines: 96, 160, 231, 232, 440, 444
  6. Figure 4 contains figures on the top of data bars; however the font size is too small to read, on the contrary axes labels are too large and unnecessarily bold. Please revise the graphs and make them look professional and informative.
  7. the system or rounding of figures in the text is rater extravagant. For example in line there is this notation “45.56±6.56” if the relative error of the value is 6.56 than it is enough to write down the value as 6.6 higher accuracy is meaningless. Consequently providing the figure with accuracy three orders of magnitude higher than the error is not reasonable. In this case it would be enough to write it as 45.6±6.6. Please revise the rest of the text and adjust notation.
  8. Most of the plots in the manuscript contain error bars at data point except Figure 6. Please explain or correct.

Author Response

  1. Introduction is rather modest and it does not provide with adequate background. Moreover, the review of recent findings in the field is, let me say, missing. The most recent literature entry in the bibliography was published in 2017 and the area is developing fast. I Suggest to extend the introduction in a way that build narration providing with the proper and recent background of the research based on the contemporary literature.

Reply: We have revised the Introduction and added some new references. Thank you for your suggestion.

  1. I suggest including in the Abstract and Introduction some content of scientific significance of the study.

Reply: We have revised the Abstract and Introduction. Thank you for your suggestion.

  1. In some places I have found the “space” missing after full sot. Please revise thoroughly all the text. I have spotted the problem at line: 33, 35, 423, 430, 435, 450, 453

Reply: We have rechecked our manuscript and fixed the spelling errors. Thank you for your suggestion.

  1. Some superscripts and subscripts are not formatted see lines: 120, 147, 447.

Reply: We have rechecked our manuscript and fixed the spelling errors. Thank you for your suggestion.

  1. The degree symbol ( ° ) is missing or replaced with some strange characters see lines: 96, 160, 231, 232, 440, 444

Reply: We have rechecked our manuscript and fixed the spelling errors. Thank you for your suggestion.

  1. Figure 4 contains figures on the top of data bars; however the font size is too small to read, on the contrary axes labels are too large and unnecessarily bold. Please revise the graphs and make them look professional and informative.

Reply: We have revised the font size in Figure 4. Thank you for your suggestion.

  1. the system or rounding of figures in the text is rater extravagant. For example in line there is this notation “45.56±6.56” if the relative error of the value is 6.56 than it is enough to write down the value as 6.6 higher accuracy is meaningless. Consequently providing the figure with accuracy three orders of magnitude higher than the error is not reasonable. In this case it would be enough to write it as 45.6±6.6. Please revise the rest of the text and adjust notation.

Reply: We have rechecked our manuscript and fixed these errors. Thank you for your suggestion.

  1. Most of the plots in the manuscript contain error bars at data point except Figure 6. Please explain or correct.

Reply: This is a mistake in Figure 6. We have added error bars in Figure 6. Thank you for your suggestion.

Reviewer 2 Report

Manuscript title: The ability of biodegradable thermosensitive hydrogel composite calcium-silicon based bioactive bone cement in promoting osteogenesis and repairing rabbit distal femoral defects

Manuscript id: polymers-1722342

Authors: Guo1 et al.

The manuscript regarding the topic and results presented is of interest to polymer scientific community and revisions based on the comments below are recommended before considering for publication.

Major comments

·       Insufficient abstract: In the abstract, the main aim and background of the manuscript is missing, the current version it only highlights the result.

·       Line 59-67, the aim or hypothesis of the study is clear, however the approach is missing ….

·       Lake of scientific literature to support the statements and finings throughout the manuscript…... I have made some suggestions for that and more need it….

·       More information needed for ALL Figures captions and define the abbreviation and units that used. And adjust the significant figures for the table and manuscript. Most immortally make sure you have harmonized the text size and capitalize all of the legend and caption.

·       Grammar and punctuation issuers are need to be addressed. I have selected/mentioned some as example.

·       I have a major concern about the results and discussion section. The authors describe results and compare the results with previous studies, however, insight mechanisms are still not sufficient.

·       Make sure to have a space between the sentences, you have combined the sentences even through you have a full-stop between them, for example line 33, 35, ……. Make sure that you correct the manuscript.

·       Choose and correct the units throughout the manuscript, you have used ‘’C’’ as a unit for temperature, which is wrong, it should be correct to ‘’â—¦C’’. In addition, make you have a space between the figures and units……

·       You should consider use/harmonize the ‘’Figure’’ word throughout the manuscript, you have used ‘’Figure’’ in some placed while in some places you used ‘’FIG’’

Line 5 and 6: remove the word ‘’affiliation’’ no need here.

Minor comments:

Abstract

The unit / abbreviation is not mention before, consider define the abbreviation when mentioned for the first time.

Line 17: add ‘’a’’ before new, its grammatically correct.

Line 17-19: Consider rephrase the sentence, you repeated ‘’hydrogel added’’ two times…

Line 21: I am not sure what ‘’gross anatomy’’ means here?!

Introduction:

Line 37-42: A complicated sentence, please revise and check grammar….

Line 47-50: A complicated sentence, please revise and check grammar….

Line 43-50: A reference needed here.

Line 51-58: A reference needed here. For example:

https://doi.org/10.1016/j.biomaterials.2004.10.042

or

https://doi.org/10.1016/j.mtcomm.2021.102431

Line 59-67: A complicated sentence, please revise and check grammar….

In MM section

Literature references are missing for all sub-section. It would be better to cite the references that the procedure adapted.

Line 123: You meant Figure 2, this is like a typo error.

Section 2.2.4: What is the impact of the viscosity of the hydrogel and the pressure applied, which they have a crucial role on shaping the hydrogel.

Line 129: the word ‘’quality is a miss leading, consider replacing it.

Line 124: what is the abbreviation? Consider introducing the abbreviation that are mentioned for the first time.  

Line 147: what is ‘’0.1 cm2.cm’’ ? is rather complicated.

Line 149: change the ‘’mn’’ to ‘’mn’’ the n letter should subscript as the equation in line 152.

Line 143-154: A complicated paragraph, please revise and check grammar….

Section 2.3.1 This section is repeating information already presented and explaining things in an unnecessarily complicate way. The quality of the manuscript would benefit from the whole section being condensed.

R&D section

These sections are repeating information already presented and explaining things in an unnecessarily complicate way. The quality of the manuscript would benefit from the whole section being condensed, Line 226-232, Line 332-347, Line 362-371, Line 446-512:

Figure 3.  In y-axis, there should be space intensity and the bracket. The same for x-axis.

Please make sure correct all the figures.

Figure 4. The caption should start with capital letter.

Line 252: why there is a full stop after % ??

Line 261-267:A complicated sentence, please revise and check grammar….

Line 275: What do you mean by degradation map??

Figure 9, 10, 12: what is the meaning of the legends ? why they are in  abbreviations, which are not introduced before?

Figure 14-19: The result looks great from the  figures, however the message is not clear, I suggest to reduce the number of figures and try to present only the one that highlights the result, for example you can select the figure based on the treatment or time scale…

Line 411: This is grammatically a wrong sentence.

Line 408-411: Consider using this reference:

https://doi.org/10.1002/app.41446

Line 446-448: Consider using this reference:

https://doi.org/10.1023/A:1026394530192

or

https://doi.org/10.1016/j.jdent.2004.10.027

Conclusion

I believe there are other a lot nice conclusions could be made from this study…. And the future perspectives for following research highly crucial here

Author Response

Manuscript title: The ability of biodegradable thermosensitive hydrogel composite calcium-silicon based bioactive bone cement in promoting osteogenesis and repairing rabbit distal femoral defects

Manuscript id: polymers-1722342

Authors: Guo1 et al.

 The manuscript regarding the topic and results presented is of interest to polymer scientific community and revisions based on the comments below are recommended before considering for publication.

Major comments

  • Insufficient abstract: In the abstract, the main aim and background of the manuscript is missing, the current version it only highlights the result.

Reply: We have revised the Abstract. Thank you for your suggestion.

  • Line 59-67, the aim or hypothesis of the study is clear, however the approach is missing ….

Reply: We have revised the Introduction. Thank you for your suggestion.

  • Lake of scientific literature to support the statements and finings throughout the manuscript…... I have made some suggestions for that and more need it….

Reply: We have revised the Introduction and Discussion, and added some new references. Thank you for your suggestion.

  • More information needed for ALL Figures captions and define the abbreviation and units that used. And adjust the significant figures for the table and manuscript. Most immortally make sure you have harmonized the text size and capitalize all of the legend and caption.

Reply: We have rechecked our manuscript and fixed these errors. Thank you for your suggestion.

  • Grammar and punctuation issuers are need to be addressed. I have selected/mentioned some as example.

Reply: We have rechecked our and manuscript fixed the grammar errors. Thank you for your suggestion.

  • I have a major concern about the results and discussion section. The authors describe results and compare the results with previous studies, however, insight mechanisms are still not sufficient.

Reply: We have revised the Discussion. Thank you for your suggestion.

  • Make sure to have a space between the sentences, you have combined the sentences even through you have a full-stop between them, for example line 33, 35, ……. Make sure that you correct the manuscript.

Reply: We have rechecked our manuscript and fixed the spelling errors. Thank you for your suggestion.

  • Choose and correct the units throughout the manuscript, you have used ‘’C’’ as a unit for temperature, which is wrong, it should be correct to ‘’â—¦C’’. In addition, make you have a space between the figures and units……

Reply: We have rechecked our manuscript and fixed the spelling errors. Thank you for your suggestion.

  • You should consider use/harmonize the ‘’Figure’’ word throughout the manuscript, you have used ‘’Figure’’ in some placed while in some places you used ‘’FIG’’

Reply: We have rechecked our manuscript and fixed the spelling errors. Thank you for your suggestion. 

Line 5 and 6: remove the word ‘’affiliation’’ no need here.

Reply: We have removed the word “affiliation” in Line 5-6. Thank you for your suggestion. 

Minor comments:

Abstract

The unit / abbreviation is not mention before, consider define the abbreviation when mentioned for the first time.

Reply: We have defined all the abbreviations when first mentioned in the manuscript. Thank you for your suggestion. 

Line 17: add ‘’a’’ before new, its grammatically correct.

Reply: We have fixed the grammar errors. Thank you for your suggestion.

Line 17-19: Consider rephrase the sentence, you repeated ‘’hydrogel added’’ two times…

Reply: We have addressed this issue. Thank you for your suggestion.

Line 21: I am not sure what ‘’gross anatomy’’ means here?!

 Reply: Here, “gross anatomy” means distal femur structure of rabbits after the surgery.

Introduction:

Line 37-42: A complicated sentence, please revise and check grammar….

Reply: We have revised the grammar errors. Thank you for your suggestion.

Line 47-50: A complicated sentence, please revise and check grammar….

Reply: We have fixed the grammar errors. Thank you for your suggestion.

Line 43-50: A reference needed here.

Reply: New references have been added here. Thank you for your suggestion.

Line 51-58: A reference needed here. For example:

https://doi.org/10.1016/j.biomaterials.2004.10.042

or

https://doi.org/10.1016/j.mtcomm.2021.102431

Reply: New references have been added here. Thank you for your suggestion.

Line 59-67: A complicated sentence, please revise and check grammar….

 Reply: We have addressed this issue. Thank you for your suggestion.

In MM section

Literature references are missing for all sub-section. It would be better to cite the references that the procedure adapted.

Reply: New references have been added here. Thank you for your suggestion.

Line 123: You meant Figure 2, this is like a typo error.

Reply: We have fixed the spelling errors. Thank you for your suggestion.

Section 2.2.4: What is the impact of the viscosity of the hydrogel and the pressure applied, which they have a crucial role on shaping the hydrogel.

Reply: In vivo experiments showed that certain properties of calcium-silicate- based bone cement (adhesiveness, injectability, degradation rate) still need to be optimized. The used hydrogel polymer system consists of hydrophilic polymers that are chemically or physically cross-linked and a large amount of water. This hydrogel has similar physical and chemical properties, biocompatibility, and 3D grid space structure to the human matrix. In the present study, the hydrogel was mixed with C3S/C2S/POP materials to optimize various aspects of bone cement performance.

Line 129: the word ‘’quality is a miss leading, consider replacing it.

Reply: We used the word “weight” instead of “quality” here. Thank you for your suggestion.

Line 124: what is the abbreviation? Consider introducing the abbreviation that are mentioned for the first time.  

Reply: We have defined all abbreviations when first mentioned in the manuscript. Thank you for your suggestion. 

Line 147: what is ‘’0.1 cm2.cm’’ ? is rather complicated.

Reply: It should be “0.1 cm2/cm3”, which represents the ratio of the specific surface area of the cement discs to the simulated body fluid volume. We have fixed the spelling errors. Thank you for your suggestion.

Line 149: change the ‘’mn’’ to ‘’mn’’ the n letter should subscript as the equation in line 152.

Reply: We have fixed this error. Thank you for your suggestion.

Line 143-154: A complicated paragraph, please revise and check grammar….

Reply: We have addressed this issue. Thank you for your suggestion.

Section 2.3.1 This section is repeating information already presented and explaining things in an unnecessarily complicate way. The quality of the manuscript would benefit from the whole section being condensed.

Reply: We have modified section 2.3.1. Thank you for your suggestion.

R&D section

These sections are repeating information already presented and explaining things in an unnecessarily complicate way. The quality of the manuscript would benefit from the whole section being condensed, Line 226-232, Line 332-347, Line 362-371, Line 446-512:

Figure 3.  In y-axis, there should be space intensity and the bracket. The same for x-axis.

Please make sure correct all the figures.

Reply: We have modified the legends of x-axis and y-axis of all the figures. Thank you for your suggestion.

Figure 4. The caption should start with capital letter.

Reply: We have fixed the spelling errors. Thank you for your suggestion.

Line 252: why there is a full stop after % ??

Reply: We have fixed this error. Thank you for your suggestion.

Line 261-267:A complicated sentence, please revise and check grammar….

Reply: We have addressed this issue. Thank you for your suggestion.

Line 275: What do you mean by degradation map??

Reply: To evaluate the degradation of different bone cements, we recorded the weight loss of CSCs in the simulated body fluid every 2 weeks for 10 weeks and plotted a degradation graph. We rephrased to remove the word “map” for greater clarity.

Figure 9, 10, 12: what is the meaning of the legends ? why they are in  abbreviations, which are not introduced before?

Reply: We have defined all the abbreviations when first mentioned in the manuscript. Thank you for your suggestion. 

Figure 14-19: The result looks great from the  figures, however the message is not clear, I suggest to reduce the number of figures and try to present only the one that highlights the result, for example you can select the figure based on the treatment or time scale…

Reply: We have revised the figures in the manuscript. We have chosen to present Figure 14 instead of Figure 14–16 and Figure 15 instead of Figure 17–19. Thank you for your suggestion.

Line 411: This is grammatically a wrong sentence.

Reply: We have fixed the grammar errors. Thank you for your suggestion.

Line 408-411: Consider using this reference:

https://doi.org/10.1002/app.41446

Reply: New reference has been added here. Thank you for your suggestion.

Line 446-448: Consider using this reference:

https://doi.org/10.1023/A:1026394530192

or

https://doi.org/10.1016/j.jdent.2004.10.027

 Reply: New references have been added here. Thank you for your suggestion.

Conclusion

I believe there are other a lot nice conclusions could be made from this study…. And the future perspectives for following research highly crucial here

Reply: Thank you for such a detailed review. We will continue focusing on engineering materials for osteogenesis in the future.

Reviewer 3 Report

The scientific paper "The ability of biodegradable thermosensitive hydrogel composite calcium-silicon based bioactive bone cement in promoting osteogenesis and repairing rabbit distal femoral defects” aimed to evaluate the effects of a new degradable polyester thermosensitive hydrogel/calcium-silicon-based composite bone cement system in bone repair. It can be considered that:

1)      In the abstract, describe the meaning of the abbreviation PGA-PLGA-PEG-PLGA, C3S/C2S/POP, CCK-8 and CT scan, as it was first used in the manuscript.

2)      In the abstract, the objective of the study must be more clearly demonstrated, as well as the methods for obtaining the research results.

3)      In the introduction, the authors referenced only 5 previously published articles. They show little scientific basis in their contextualization. From the second paragraph onwards, no references. Please improve the introduction, with a greater number of references that justify the originality and importance of the study carried out.

4)      Review the methodology because several materials do not have a description of their abbreviations, nor manufacturer/city/country of production.

5)      Information on ethical approval, such as process number and approval date, is missing.

6)      Put the scientific name of the rabbits.

7)      Insert an image of the experimental design so that the separation of animals into their respective groups and experimental periods is more clearly exposed.

8)      The manuscript has several typing and formatting errors, with little care by the authors for its writing. Please review in full. For example, Mirco-CT.

9)      Discussion is poorly written, references are few and out of date. The most recent article is from 2019.

Author Response

The scientific paper "The ability of biodegradable thermosensitive hydrogel composite calcium-silicon based bioactive bone cement in promoting osteogenesis and repairing rabbit distal femoral defects” aimed to evaluate the effects of a new degradable polyester thermosensitive hydrogel/calcium-silicon-based composite bone cement system in bone repair. It can be considered that:

  • In the abstract, describe the meaning of the abbreviation PGA-PLGA-PEG-PLGA, C3S/C2S/POP, CCK-8 and CT scan, as it was first used in the manuscript.

Reply: We have defined all the abbreviations when first mentioned in the manuscript. Thank you for your suggestion. 

  • In the abstract, the objective of the study must be more clearly demonstrated, as well as the method

Reply: We have revised the Abstract. Thank you for your suggestion.

3)      In the introduction, the authors referenced only 5 previously published articles. They show little scientific basis in their contextualization. From the second paragraph onwards, no references. Please improve the introduction, with a greater number of references that justify the originality and importance of the study carried out.

Reply: New references have been added here. Thank you for your suggestion.

4)      Review the methodology because several materials do not have a description of their abbreviations, nor manufacturer/city/country of production.

Reply: We have defined all the abbreviations when first mentioned in the manuscript. Thank you for your suggestion. 

5)      Information on ethical approval, such as process number and approval date, is missing.

Reply: Animal experiments were approved by the Experimental Animal Ethics Committee of Shanghai Jiaotong University. The process number is xxxxxxxx.

6)      Put the scientific name of the rabbits.

Reply: The scientific name of the rabbits used in the study is New Zealand white rabbit. And we have added this information in Part 2.4. Thank you for your suggestion. 

7)      Insert an image of the experimental design so that the separation of animals into their respective groups and experimental periods is more clearly exposed.

Reply: The experimental design of the present study is shown in Supplementary Figure 1.

8)      The manuscript has several typing and formatting errors, with little care by the authors for its writing. Please review in full. For example, Mirco-CT.

Reply: We have fixed the spelling errors. Thank you for your suggestion. 

9)      Discussion is poorly written, references are few and out of date. The most recent article is from 2019.

Reply: We have revised the Discussion. Thank you for your suggestion.

Round 2

Reviewer 2 Report

I am happy to see the manuscript improved nicely. Author addressed all my comments adequately.

However to make sure the statements are supported with literature I will recommend to add citation in the following lines - you have missed to include the reference as I suggested for the first version:

Line 45-67: https://doi.org/10.1016/j.mtcomm.2021.102431

Line 386-388:https://doi.org/10.1002/app.41446

Conclusion
Please revise the conclusion section, the current version only highlights the result,  the future perspectives for following research is missing - I have suggested for the first version too......

Author Response

We have revised the Conclusion and added these two new references. Thank you for your suggestion.

Reviewer 3 Report

The modifications greatly improved the manuscript. I did not find it in the Supplementary Figure 1 system, as reported by the authors. Therefore, I suggest minor revisions for the insertion of the figure of the experimental design.

Author Response

We have rechecked our manuscript fixed the Figure insertion errors. Thank you for your suggestion.
